# Magnetic Domain Structure of $Lu_{2.1}Bi_{0.9}Fe_5O_{12}$ Epitaxial Films Studied by Magnetic Force Microscopy and Optical Second Harmonic Generation

**Marina Temiryazeva** [1,2,†] ⬤, **Evgeny Mamonov** [3,†], **Anton Maydykovskiy** [3,†], **Alexei Temiryazev** [1,†] ⬤
and **Tatiana Murzina** [3,*] ⬤

1   Kotel'nikov Institute of Radioengineering and Electronics of RAS, Fryazino Branch, Vvedensky Sq. 1, Fryazino 141190, Russia
2   Kotel'nikov Institute of Radioengineering and Electronics of RAS, Moscow 125009, Russia
3   Department of Physics, Moscow State University, Moscow 119991, Russia
*   Correspondence: murzina@mail.ru
†   These authors contributed equally to this work.

**Abstract:** Magnetic structure of functional magnetic dielectrics is traditionally of high interest. Here, we use the magnetic force microscopy (MFM) and nonlinear-optical probe of second harmonic generation for studies of surface domain structure of monocrystalline $Lu_{2.1}Bi_{0.9}Fe_5O_{12}$ garnet films. The transformation of the magnetic domains under the application of the dc magnetic field is revealed by the MFM for both the top-view and the cleavage of the iron-garnet layer. Complementary magnetic force and second harmonic generation microscopy show that the considered film reveals the magnetization inclined with respect to the film's normal, with its orientation being inhomogeneous within the film's thickness. The second harmonic generation (SHG) microscopy confirms the zigzag structure of the surface-closing domain with the magnetization containing in-plane and out-of-plane magnetization components. We believe that these features of magnetic behavior of garnet films are important for the design of garnet-based magnetic devices.

**Keywords:** magnetic force microscopy; garnet films; magnetic domain structure; optical second harmonic generation

## 1. Introduction

Physics of magnetic domains is a fascinating research topic, as magnetic ordering provides rather complicated and beautiful structures [1,2]. One of the most studied objects here are iron-garnet single crystals; their domain structure is being studied for quite a time by different methods, including the so-called powder and optical polarization techniques [1,3–6]. It is recognized that the organization of domains is affected to a great extent by the imperfections of the structure, such as dislocations, as well as the substrate and interface effects where mechanical stress can be substantially important [7–10]. In part, these studies were performed using the nonlinear optical technique of optical second harmonic generation (SHG), which is known for its high sensitivity to the main properties of surfaces and interfaces, including magnetic ones [11–15]. It is also well recognized that this method provides unique possibilities for the studies of biological objects and tissues [16–19]. As compared to commonly used magneto-optical methods, magnetization-induced SHG effects consisting in the polarization plane rotation, phase or intensity variations, are one or two orders of magnitude higher, which is beneficial for the studies of magnetic nanostructures with small total magnetic moment [20,21] and allowed to visualize magnetic behavior of magnetic films with the thickness down to a few monolayers [22].

Magnetic force microscopy (MFM) [23,24] makes it possible to visualize the distribution of magnetic field on the surface of magnetic materials. The stray fields mainly

reflect the near-surface distribution of magnetization. In the case of transparent garnet films, the results of MFM can be compared with linear magneto-optical studies, which provide information on the magnetic properties of the bulk of the films. It was shown that the structure of magnetic domains observed by the MFM is rather complicated and differs substantially from that of the bulk ones. This was associated with the presence of surface domains that do not penetrate through the bulk of the films and reveal a fractal-like structure located above the bulk stripe domains [25,26].

Quite recently, we started the complex studies of magnetic properties of surface layers of garnet films by a combination of two surface-sensitive MFM and SHG techniques. The experiments were focused on the investigation of the residual surface domain structure as well as its variation under the application of an in-plane DC magnetic field. The main result is that in the case of (111) $Lu_{2.1}Bi_{0.9}Fe_5O_{12}$ film, a zigzag-like ordering of surface domains exists with partially in-plane magnetization. Here, we develop this approach and perform the MFM studies of modifications of the surface domain structure of a similar garnet film under the application of an in-plane and out-of-plane DC magnetic field. In combination with the second harmonic generation studies, the obtained results show that the stripe magnetic domains possess an inclined magnetization direction, as well as inhomogeneity in its orientation along the films' normal.

## 2. Experimental Procedure and Samples under Study

### 2.1. Experimental Setups

Magnetic force microscopy studies were performed using a SmartSPM atomic force microscope manufactured by AIST-NT (currently produced by HORIBA Scientific). In order to study the tiny domain structure of garnet films, we used a special type of tip covered by an 8 nm thick Co/Pt multilayered film with perpendicular magnetic anisotropy [27], which ensures a low magnetic moment of the probe. This made it possible to perform MFM measurements without distorting the magnetic structure of the film even at a small tip-sample distance, which provides good spatial resolution. In order to exclude the influence of the film morphology on the magnetic image of the structure, the measurements were carried out by a standard two-pass technique (lift mode). At the first pass, the topography was recorded in the tapping mode for each line, and at the second pass, the magnetic response was registered for a fixed distance between the probe and the surface.

The phase shift of the probe oscillations was used as a signal for the MFM image. The setting of the phase detector in our device is such that as the resonant frequency of the probe decreases, the phase of the signal increases. Thus, the light areas on the MFM images correspond to the attraction of the probe to the sample. We studied the modification of the magnetic domains under the application of the in-plane and out-of-plane DC magnetic field.

The MFM technique was applied for the studies of magnetic field induced changes in the domain structure of the garnet film. These measurements were carried out using the magnets built into the SmartSPM. When the magnetic field changed, a series of MFM images consisting of several hundreds of scans were taken, from which a movie was further mounted. The films are presented in Supplementary Materials. The most characteristic frames were chosen from these movies and shown as the figures in this paper.

Nonlinear-optical microscopy of garnet films was performed using the setup described in detail in [28]. In brief, the radiation of a Ti-sapphire laser (at the wavelength of 850 nm, with the pulse duration 60 fs and the mean power of 50 mW) was focused on the sample by the Mitutoyo M Plan Apo 100× objective with the NA = 0.7. Transmitted radiation at the second harmonic wavelength was collected by a similar objective, passed through the necessary set of filters and an analyzer, and was detected by the photomultiplier Hamamatsu R4220p.

### 2.2. Samples

The studied samples consist of a 10 μm thick monocrystalline $Lu_{2.1}Bi_{0.9}Fe_5O_{12}$ film made by the liquid phase epitaxy on (111) $Gd_3Ga_5O_{12}$ substrate. This film is characterized

by high absorption in the wavelength range below ≈550 nm, which is typical for garnet films [29]. This means that in the nonlinear-optical experiments, the film is highly transparent at the fundamental wavelength of 850 nm, while the central SHG wavelength at 425 nm falls in the absorption band. When taking into account the absorption coefficient of typical monocrystalline garnets [29], the subsurface layer corresponding to the escape length at the SHG wavelength, which forms nonlinear response, is about 2 μm. This provides an instrument for testing the relatively thin surface garnet layer.

The magneto-optical Kerr effect in the transversal or polar geometries reveals that the in-plane saturating field is less than 1 kOe, while the out-of-plane exceeds 3 kOe.

## 3. Results and Discussion

### 3.1. Magnetic Force Microscopy

Figure 1 shows a series of MFM images obtained by remagnetization of the film in an in-plane magnetic field. More details can be found in the movie (see the movie S1–*MFM–Garnet–in–lateral–H.avi* in Supplementary Materials). In this experiment, the film orientation was chosen in such a way that initially the domains in the scanning area were oriented at an angle to the direction of the external field (oriented horizontally in the figure). One can see a biperiodic domain structure with a fundamental period of about 4 μm and a modulation period of 1.6 μm. As the magnetic field increases, the stripe domains turn (Figure 1a–c), and the character of the longitudinal modulation (Figure 1d–f) changes. Further, the longitudinal modulation first disappears (Figure 1g), and then, the strip domain structure itself (Figure 1h), so the film turns into the saturated state. If we now slightly reduce the external field, the domain structure appears again, while already oriented in the direction of the applied magnetic field (Figure 1i).

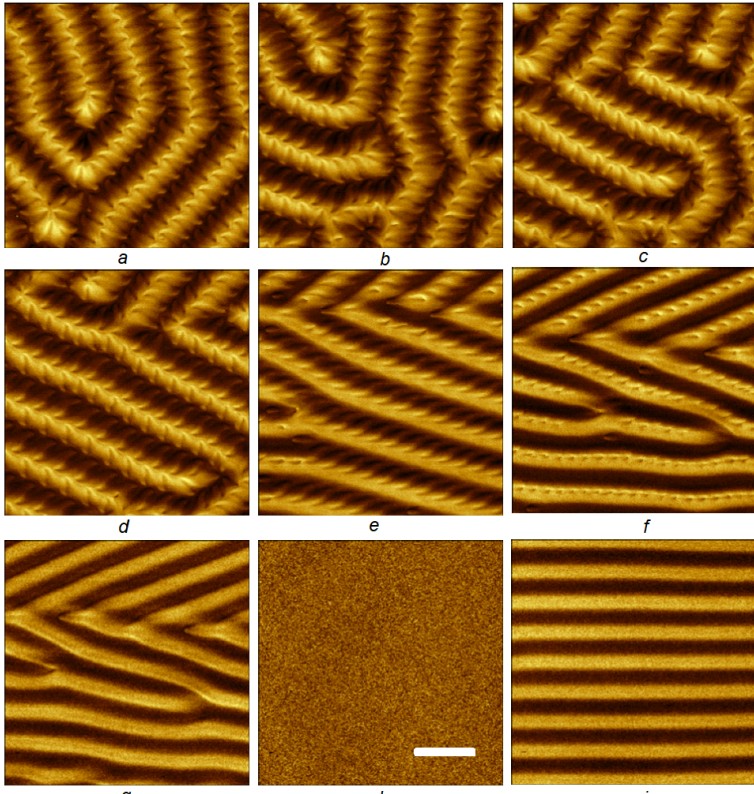

**Figure 1.** A series of MFM images of the garnet film in the in-plane magnetic field. The field is directed along the X axis; the film was initially in the demagnetized state. (**a**) H = 0 Oe, (**b**) H = 35 Oe, (**c**) H = 40 Oe, (**d**) H = 65 Oe, (**e**) H = 205 Oe, (**f**) H = 260 Oe, (**g**) H = 370 Oe, (**h**) H = 470 Oe, (**i**) H = 370 Oe. The scale bar corresponds to 5 μm.

In the next series of experiments, we considered in more detail the process of the film magnetization reversal for this domain orientation. With a step of 1 Oe, a series of MFM images was taken while changing the field in the range from −400 Oe to 400 Oe (see movie in Supplementary Materials, video S2–*MFM–Garnet–in–lateral–H.avi*). Figure 2 shows some of the acquired scans. First, a stripe domain structure (DS) appears (Figure 2a). Further, a dot structure with a period of 1 μm and a minimum size of an individual element of about 200 nm is formed inside the dark domains (Figure 2b). We have increased the contrast in this figure to show the presence of such a periodic structure. With the further increase of the field, after a series of transformations (Figure 2c), the structure with a pronounced periodic structure along the stripe domain is formed.

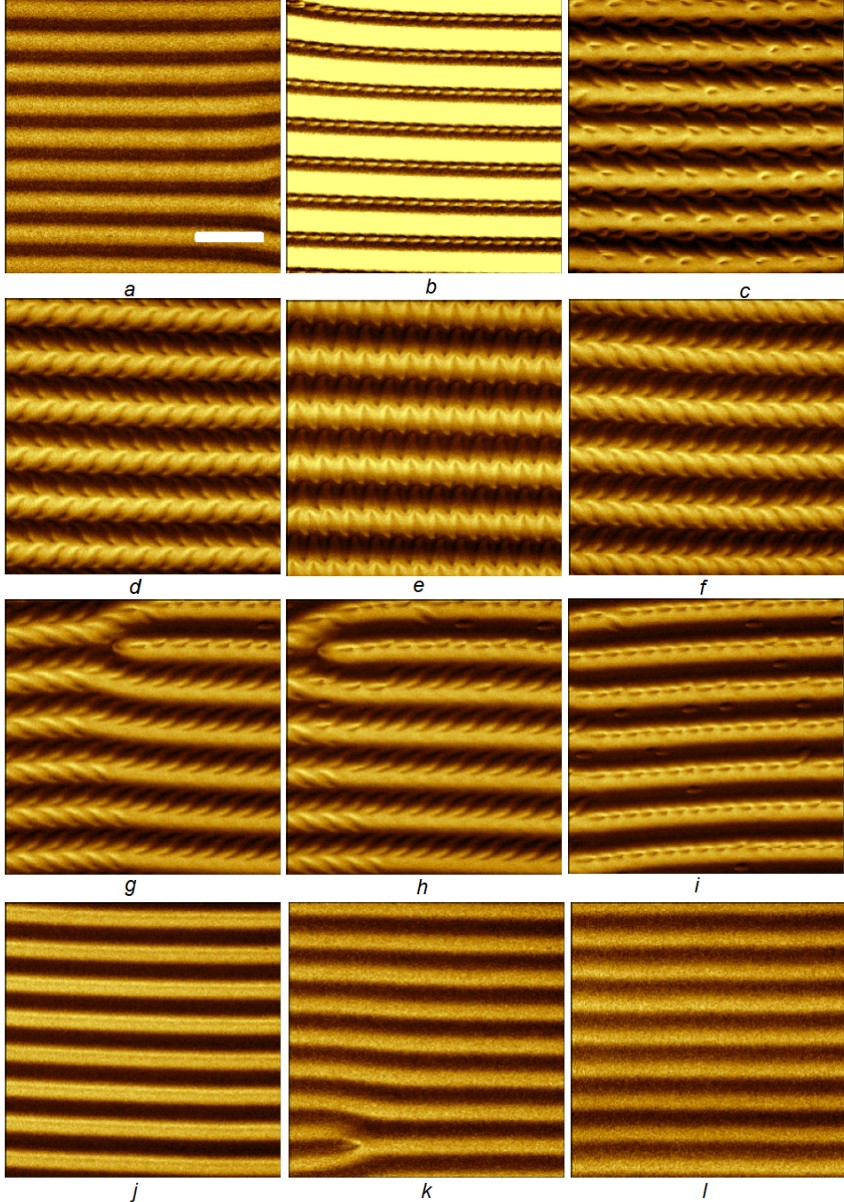

**Figure 2.** MFM images of the garnet film in the in-plane magnetic field. The film was initially saturated. (**a**) H = −387 Oe, (**b**) H = −235 Oe, (**c**) H = −142 Oe, (**d**) H = −65 Oe, (**e**) H = 0 Oe, (**f**) H = 65 Oe, (**g**) H = 176 Oe, (**h**) H = 178 Oe, (**i**) H = 192 Oe, (**j**) H = 300 Oe, (**k**) H = 394 Oe, (**l**) H = 410 Oe. The scale bar corresponds to 5 μm.

When the sign of the external field is reversed, the angle of the protruding wedges changes (Figure 2d–f). It should be noted that when watching movies, one can sometimes see a jitter of the picture—a shift of domains up and down (in the vertical direction). This is not related to errors in the positioning of the scanning field; it is constant throughout the entire series with an accuracy of 10–20 nm. The magnetization reversal is associated with the motion of defects in the stripe domain structure, e.g., magnetic dislocations [30]. This can be observed most clearly over a larger area during magneto-optical registration based on the Faraday effect (see the movie *Magnetooptics–Garnet–in–lateral–H.avi* in Supplementary Materials). Sometimes, the movement of a dislocation (Figure 2g,h) enters the MFM scan area. It causes deformation of the stripe DS and vertical displacement of the neighboring domains. In the case when a dislocation passes outside the scanning region, we see only a shift in the domain structure. Note that in the case shown in Figure 2, the motion of a magnetic dislocation is associated with a change in the type of longitudinal modulation from two-sided (Figure 2g) to one-sided, and then to point structure (Figure 2h,i). Each of these transformations is accompanied by a decrease in the period of the stripe DS. At zero external magnetic field, the period is 3.84 µm, while in a field of 410 Oe, the period is 1.6 times smaller and is about 2.39 µm.

The next series of experiments was carried out with the external magnetic field applied perpendicular to the plane of the film (see the movie in Supplementary Materials, file *MFM–Garnet–in–vertical–H.avi*). Starting from saturating fields, we gradually reduced the field until, at the field strength of about 600 Oe, weak domain outlines appeared on the MFM images (Figure 3a). At this point, the magneto-optical images (not presented in this paper) showed quite clearly that the film contained a lattice of round domains. Probably, they were located in the internal volume of the film and did not create significant stray fields recorded by the MFM. As the field decreased, the domain walls came out to the film surface, the MFM image became more clear, and the domains were extended along the surface (Figure 3b). Further, an additional fine dot pattern (Figure 3c) appeared; the domains expanded and acquired an internal structure (Figure 3d). In small fields, an array of separate domains of complex shape was formed. As can be seen from the comparison of Figures 1–3, the type of DS in the zero field depends significantly on the prehistory.

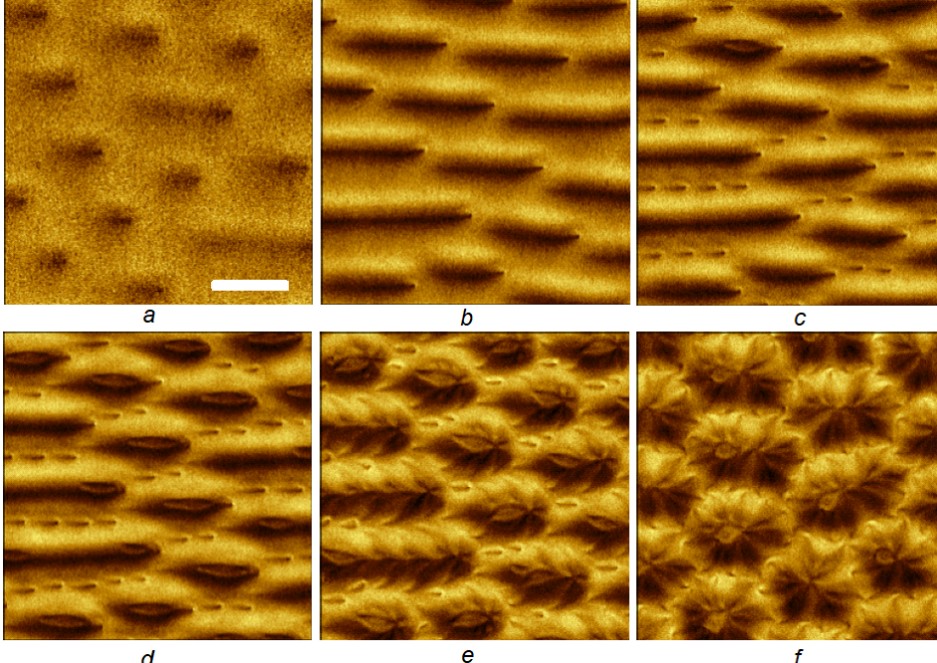

**Figure 3.** MFM images of the garnet film in the out-of-plane magnetic field. (**a**) H = 577 Oe, (**b**) H = 427 Oe, (**c**) H = 358 Oe, (**d**) H = 296 Oe, (**e**) H = 120 Oe, (**f**) H = 0 Oe. The scale bar corresponds to 5 µm.

It was noted in [25,26] that during the MFM measurements of iron-garnet films with a thickness of several microns, the stray fields, the presence of which is recorded by MFM, are formed mainly by the magnetization structure in a narrow near-surface layer at the film–air interface. This explains the observation of submicron domains in a film of 10 μm thick. Their lateral dimensions are apparently determined by the thickness of the layer in which they are localized. In the internal volume of the film, the distribution of magnetization can differ significantly from the surface one. This can be demonstrated by carrying out the MFM measurements on a cleavage (end face) of the film [31].

We performed similar measurements in an external magnetic field directed perpendicular to the film surface (see the movie *MFM–Garnet–cleavage.avi* in Supplementary Materials). Figure 4 shows some of the obtained MFM images. The lines indicate the approximate positions of the boundaries of the magnetic film with the substrate placed on the right and air on the left from the garnet layer. At the zero field, on the film side adjacent to the boundary with air, light and dark domains are seen, inclined with respect to the surface normal (Figure 4a). As the field increases, the domains adjacent to the boundary with the substrate (Figure 4b) also become clearly visible. At higher fields, one can observe domains located at some distance from the film/air interface (Figure 4c), which correlate with the results of the previous experiment. The scan obtained after the magnetic field was again turned to zero (d) shows that the period of the domain structure has changed, branching of one of the domains is observed, and a separate near-surface domain has appeared at the boundary with air in the region of this branching.

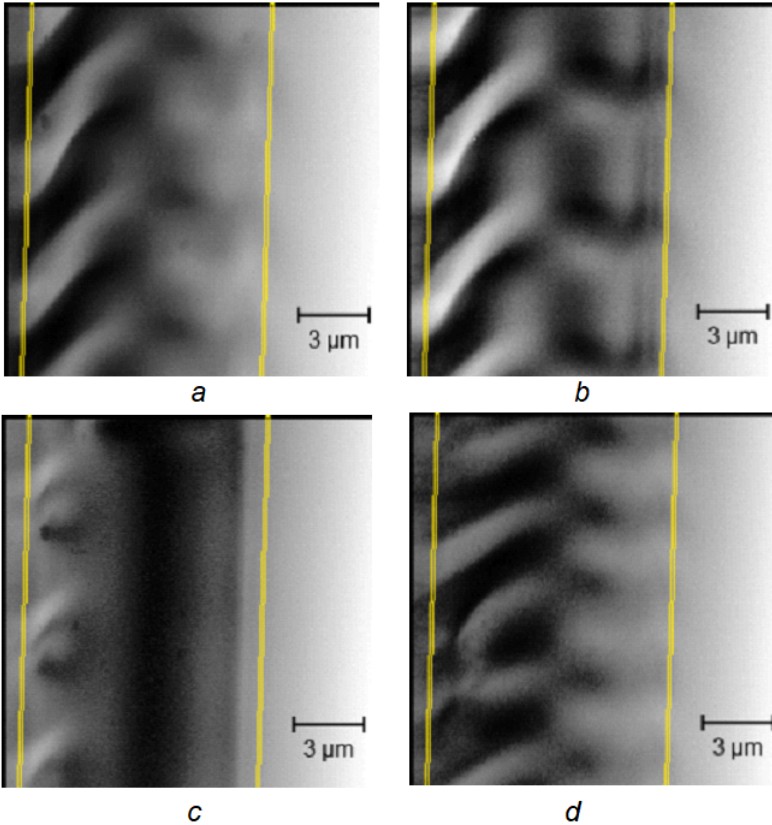

**Figure 4.** MFM images on a cleavage (end face) of the garnet film in the external out-of-plane magnetic field (**a**) H = 0 Oe, (**b**) H = 190 Oe, (**c**) H = 560 Oe, (**d**) H = 0 Oe.

It should be noted that a significant difference in the structure of domains adjacent to different interfaces may be due to the inhomogeneity of the magnetic properties of the film with the thickness. This inhomogeneity can be estimated from the spin-wave resonance spectrum [32,33]. The measurements performed for this film showed that the

value $4\pi M\text{-}H_{eff}$, where $M$ is the saturation magnetization and $H_{eff}$ is the effective field of uniaxial anisotropy, lies in the range from 1150 to 1420 Oe, so the difference is about 270 Oe.

### 3.2. Second Harmonic Generation Microscopy

Configuration of the surface magnetic domains in the absence of external DC magnetic field was also studied by nonlinear-optical magnetic microscopy that involved the surface-sensitive polarization-resolved SHG microscopy technique [34–36]. This surface sensitivity is intrinsic to the SHG probe in the case of centrosymmetric materials, e.g., pure iron-garnet films with cubic crystallographic structure. At the same time, garnet films doped by heavy ions possess slightly distorted structure; so, the layer with the thickness of about 2 μm comparable to the escape length at the SHG wavelength contributes to the nonlinear optical signal.

Figure 5a shows the SHG microscopy image of the studied garnet film with the stripe domains oriented in the vertical direction. The excitation of the film was performed by linearly polarized radiation of the Ti-sap laser with the polarization plane perpendicular to the stripe domains, and the SHG radiation with similar polarization was detected, as shown by the arrows. One can see that the SHG pattern reveals two periods: (i) of approximately 4 μm along the $X$ axis that corresponds to the stripe domains, and (ii) of about 1.6 μm along the Y axis with a zigzag shape; the coordinate frame is indicated in the figure. It is worth noting the contrast in the SHG pattern appears due to high values of the magnetization-induced modulation of the SHG intensity intrinsic to this method [37], in which the amplitude of the magnetization-induced contribution of the SHG field $E_{2\omega}^{M}$ that is odd with respect to the magnetization **M**, can be of the same order of magnitude as the nonmagnetic (crystallographic) one, $E_{2\omega}^{cr}$. As a result, due to the presence of residual magnetization, the SHG intensity is given by the expression $I_{2\omega}=(\mathbf{E}_{2\omega}^{cr} + \mathbf{E}_{2\omega}^{M})^2$. Changing the direction of **M** leads to the appearance of the magnetic contrast of the SHG intensity, $\rho_{2\omega} \propto E_{2\omega}^{M}/E_{2\omega}^{cr}$, which can reach the values of dozens of percents. Such an effect allows to figure out the residual magnetic domain structure of the studied garnet film.

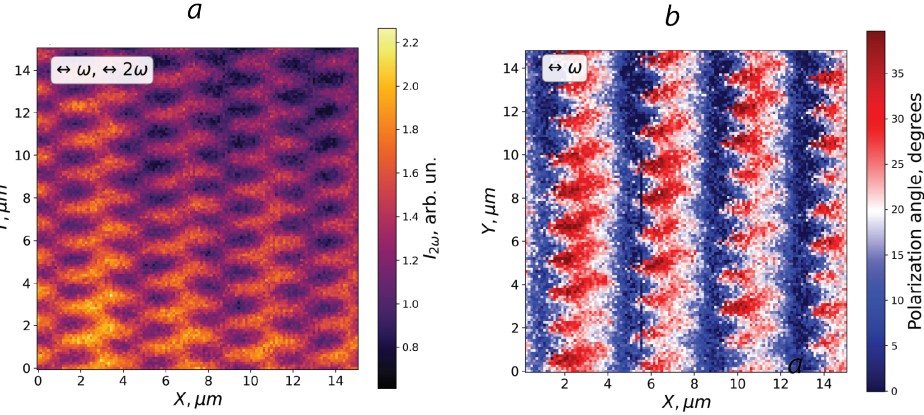

**Figure 5.** (**a**) SHG microscopy image of the garnet film measured for the parallel polarizations of the fundamental beam and SHG that are perpendicular to the domains; (**b**) SHG polarization-plane rotation spatial distribution.

It is worth noting that in the simplest approximation when we consider only the largest, magnetization-induced SHG term linear in **M**, only $M_y$ in our experimental geometry can contribute to the SHG signal, while another in-plane component $M_x$ as well as that oriented perpendicularly to the film's surface ($M_z$) do not appear in the SHG response due to symmetry arguments [12]. Thus, the pattern in the SHG intensity shown in Figure 5 indicates that in the surface layer of the garnet film, a periodic modulation of $M_y$ is observed. Interestingly, the periods of the modulated surface magnetic structure obtained by the MFM and SHG microscopy are the same within the experimental accuracy. We can conclude that in the absence of an external magnetic field, there is a surface regular domain



structure above the stripe domains, with the magnetization inclined with respect to the film surface and with the in-plane component mostly parallel to the stripes.

Figure 5b shows the pattern of the SHG polarization plane rotation relative to the polarization of the fundamental wave in the garnet film with residual domain structure. One can see that the adjacent domains rotate the SHG polarization plane in the opposite directions. This may be consistent with the opposite orientation of **M** in them, while the zigzag modulation can be attributed to the contribution of surface domains with the $M_y$ component of the magnetization with the opposite signs in the adjacent domains.

### 4. Conclusions

Summing up, we present the experimental studies of the surface domain structure of epitaxial $Lu_{2.1}Bi_{0.9}Fe_5O_{12}$ garnet films. Two methods that are specifically sensitive to magnetic properties of the surfaces of magnetics, namely magnetic force microscopy and optical second harmonic generation, were used. We demonstrate that in addition to the stripe domain structure of 10 µm thick garnet films, surface periodic modulation of the magnetic structure appears associated with the connecting domains. Combined analysis of the MFM and SHG data show that the residual magnetization of these domains contains both in-plane and out-of-plane components. MFM probe is applied for the studies of the modification of the surface domains under the application of the DC magnetic field of various orientations. MFM images reveal the appearance of fractal-like residual surface domains after the application of the out-of-plane magnetic field.

**Supplementary Materials:** The following supporting information can be downloaded at: https:// www.mdpi.com/article/10.3390/magnetochemistry8120180/s1, Video S1: 1_MFM_Garnet_in_lateral_ H.avi; Video S2: 2_MFM_Garnet_in_lateral_H.avi; Video S3: Magnetooptics_Garnet_in_lateral_H.m4v; Video S4: MFM_Garnet in vertical H.avi; Video S5: MFM_Garnet_cleavage.avi.

**Author Contributions:** Conceptualization, A.T., E.M., T.M.; methodology, M.T., E.M.; software, A.T., E.M.; validation, M.T., T.M. and A.M.; formal analysis, A.T., E.M.; investigation, M.T., E.M., A.T.; resources, A.M.; data curation, M.T., A.M.; writing—original draft preparation, T.M., A.T.; writing—review and editing, A.T., E.M.; visualization, A.T.; supervision, T.M.; project administration, A.T., M.T.; funding acquisition, T.M. All authors have read and agreed to the published version of the manuscript.

**Funding:** This research was funded by Russian Science Foundation, Grant 19-72-20103. MFM studies were supported by Russian Science Foundation, Grant 19-19-00607-P.

**Institutional Review Board Statement:** Not applicable.

**Informed Consent Statement:** Not applicable.

**Data Availability Statement:** The data presented in this paper are available in the Supplementary materials.

**Acknowledgments:** The authors acknowledge useful discussions with M. Logunov.

**Conflicts of Interest:** The authors declare no conflict of interest.

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
