# Peer review of "Magnetic Domain Structure of Lu2.1Bi0.9Fe5O12 Epitaxial Films Studied by Magnetic Force Microscopy and Optical Second Harmonic Generation"

_magnetochemistry, doi:10.3390/magnetochemistry8120180_

Round 1
Reviewer 1 Report
Dear Editor
The manuscript entitled "Magnetic domain structure of (BiLu)3Fe5O12 epitaxial films studied by magnetic force microscopy and optical second harmonic generations" is well written, in an easy-to-follow manner. The work reported in this contribution would be of great interest to the reader of Molecular Catalysis. The precise description
of each characterization method is appreciable. I would strongly
recommend this work for publication in this journal.
Regards
Author Response
We are very thankful to the Reviewer for high high evaluation of our work!!
Reviewer 2 Report
Marina Temiryazeva et al. reported the surface domain structure of a monocrystaline Lu2.1Bi0.9Fe5O12 garnet film using magnetic force microscopy and a nonlinear-optical probe of second harmonic generation in this article. Their experimental findings support the surface domain's zigzag topology and the presence of both in-plane and out-of-plane magnetization components. The experimental findings that are reported in the main text and supporting materials are strong, trustworthy, and physically significant. I think that the results of this study could influence other scientists. I would gladly recommend that this paper be published. There are numerous typos, including sigzag, inperfections and other errors. Correct them, please.

Author Response
The authors are very thankful to the Reviewer for high evaluation of our manuscript and results presented therein.
Author Response
We are thankful to Reviewer for positive opinion of our manuscript and useful comments that were made. Below we answer these comments point by point; the comments made by the Reviewers are written in Italic.
- “The author mention the composition of their system (BiLu)3Fe5O12 in title, while in abstract it is Lu2.1Bi0.9Fe5O12. The author should change Lu2.1Bi0.9Fe5O12 everywhere in the manuscript. “
- we apologize for this inaccuracy. Everywhere in the text and in the title we corrected the composition of the garnet to Lu2.1Bi0.9Fe5O12.
- “Sigzag, spelling mistake, author need to do check properly the spelling and grammatical mistakes.”
- the typos were corrected. Sorry.
- “The author wrongly cite for the description “Initially much information was obtained by optical polarization method and powder technique” [2]. It not clear how which powder or polarization method technique determined the magnetic domain structure.”
- The article [2] is an example of using both powder and polarization methods to visualize domain structure. Nevertheless, a reference to a book describing these methods was added. We also modified the sentence; now it sounds “One of the most studied objects here are iron garnet single crystals; their domain structure is being studied for quite a time by different experimental methods, including the so called powder and optical polarization techniques”
- “The author confirm the existence of magnetic domain structure by MFM and SHG techniques. But don’ t analyze the data, why zigzag domain evolves with both in-plane and out-of-plane mangetic field.”
- We are thankful to the Reviewer for this concern. We came to the conclusion that the zigzag domains, which are associated with the surface closure domains, possess both components of the magnetization, as (i) they appear in the MFM measurements and thus should be attributed to the out-of-plane component of magnetization (this is marked in the manuscript), and (ii) they appear in the similar manner in the SHG measurements. Based on the symmetry analysis, the SHG maps for the parallel or orthogonal polarizations of the fundamental beam and of the SHG wave (at normal incidence, which is the case of the SHG microscopy studies) can appear only due to the existence of the lateral components of magnetization on the film surface. These in-plane components of M as it is indicated in the text (3rd paragraph in Section 3.2, “Second harmonic generation microscopy”).